# Architecture and activation of human muscle phosphorylase kinase

Xiaoke Yang[1,5], Mingqi Zhu[1,5], Xue Lu[2,3], Yuxin Wang [1] & Junyu Xiao [1,2,4] ✉

The study of phosphorylase kinase (PhK)-regulated glycogen metabolism has contributed to the fundamental understanding of protein phosphorylation; however, the molecular mechanism of PhK remains poorly understood. Here we present the high-resolution cryo-electron microscopy structures of human muscle PhK. The 1.3-megadalton PhK $\alpha_4\beta_4\gamma_4\delta_4$ hexadecamer consists of a tetramer of tetramer, wherein four $\alpha\beta\gamma\delta$ modules are connected by the central $\beta_4$ scaffold. The $\alpha$- and $\beta$-subunits possess glucoamylase-like domains, but exhibit no detectable enzyme activities. The $\alpha$-subunit serves as a bridge between the $\beta$-subunit and the $\gamma\delta$ subcomplex, and facilitates the $\gamma$-subunit to adopt an autoinhibited state. $Ca^{2+}$-free calmodulin ($\delta$-subunit) binds to the $\gamma$-subunit in a compact conformation. Upon binding of $Ca^{2+}$, a conformational change occurs, allowing for the de-inhibition of the $\gamma$-subunit through a spring-loaded mechanism. We also reveal an ADP-binding pocket in the $\beta$-subunit, which plays a role in allosterically enhancing PhK activity. These results provide molecular insights of this important kinase complex.

Reversible protein phosphorylation, mediated by the interplay between >500 kinases and >100 phosphatases in human, is of central importance in physiology and disease[1–3]. This fundamental regulatory mechanism is discovered by Fischer and Krebs in their studies of glycogen metabolism[4,5]. Glycogen is the storage form of glucose in animals, and glycogen phosphorylase (GP) catalyzes the rate-limiting step in glycogenolysis to break down glycogen into glucose. Phosphorylase kinase (PhK) phosphorylates GP and transforms it from the inactive *b* form to the active *a* form to initiate this important biochemical process. The discovery of this phosphorylation-mediated regulation mechanism and the identification of PhK were major milestones in biochemistry and cell biology, as they not only provided crucial insights into the energy production process in human, but also revolutionized our understanding of cellular signaling events.

PhK is one of the largest and most complex protein kinases. It contains four subunits: $\alpha$, $\beta$, $\gamma$, and $\delta$, which assemble into a 1.3-megadalton $\alpha_4\beta_4\gamma_4\delta_4$ hexadecamer with a butterfly-like appearance[6,7]. The $\alpha$- and $\beta$-subunits are >1000 residues large proteins, and together account for >80% of the molecular weight of the

PhK complex[8]. However, limited information is available regarding their 3D structures[9,10]. The $\delta$-subunit was identified as calmodulin[11,12]. Unlike the interactions between calmodulin and many other calmodulin-binding proteins, however, calmodulin binds tightly regardless of the presence of $Ca^{2+}$ and serves as an integral component of the PhK holoenzyme. The $\gamma$-subunit bears the catalytic activity and consists of an N-terminal kinase domain (KD) and a C-terminal regulatory domain (CRD). The crystal structure of the rabbit muscle PhK $\gamma$-subunit (rabbit PhK$\gamma$) KD reveals a canonical protein kinase fold that resembles the cAMP-dependent protein kinase (PKA)[13,14]. However, a notable difference is the presence of a highly conserved Glu182 in the rabbit PhK$\gamma$, which replaces Thr197 in the activation segment of PKA. This Glu182 forms an ion pair with Arg148, located before the catalytic Asp149. Consequently, the rabbit PhK$\gamma$ KD adopts a constitutively active conformation without requiring activation through phosphorylation. The CRD contains calmodulin-binding motifs[15]. The truncated $\gamma$-subunit without CRD is constitutively active[16–18], suggesting that CRD inhibits KD's activity. $Ca^{2+}$ activates PhK activity, presumably by

[1]State Key Laboratory of Protein and Plant Gene Research, School of Life Sciences, Peking University, Beijing, P.R. China. [2]Changping Laboratory, Beijing, P.R. China. [3]Academy for Advanced Interdisciplinary Studies, Peking University, Beijing, P.R. China. [4]Peking-Tsinghua Center for Life Sciences, Peking University, Beijing, P.R. China. [5]These authors contributed equally: Xiaoke Yang, Mingqi Zhu. ✉e-mail: junyuxiao@pku.edu.cn

binding to calmodulin and causing conformational changes; however, the molecular mechanism remains insufficiently understood.

Here we investigate the molecular architecture and activation mechanism of PhK. We determine a high-resolution cryo-electron microscopy (cryo-EM) structure of human muscle PhK in the inactive state, which elucidates the architecture and subunit organization of the PhK hexadecamer. The C-terminal region of the γ-subunit CRD docks onto the α-subunit and inhibits KD using a pseudo-substrate mechanism. Ca²⁺-free calmodulin is attached to the γ-subunit in a compact conformation and interacts with both the N-terminal region of CRD and KD. We have further studied the cryo-EM structure of PhK in the presence of Ca²⁺, and propose a spring-loaded mechanism of how Ca²⁺-induced conformational change of calmodulin de-inhibits the kinase activity.

## Results

### Overall structure of the PhK holoenzyme

The four subunits of human muscle PhK were co-expressed in HEK293F cells, and the resulting enzyme complex was isolated with a high degree of purification (Fig. 1a, b). This recombinant PhK exhibited a robust Ca²⁺-dependent kinase activity towards GP (Fig. 1c), which was similar to that of PhK purified from rabbit skeletal muscle[19,20]. The structure of the human muscle PhK in its inactive state, prepared in the presence of the chelating agent ethylene glycol-bis(β-aminoethylether)-tetraacetic acid (EGTA), was determined using the single-particle cryo-EM method, with an overall resolution of 2.9 Å (Supplementary Fig. 1, Table 1). The density of one half of the complex is superior to that of the other half. Local refinements were further performed for the αβγδ and γδ subcomplexes that display good

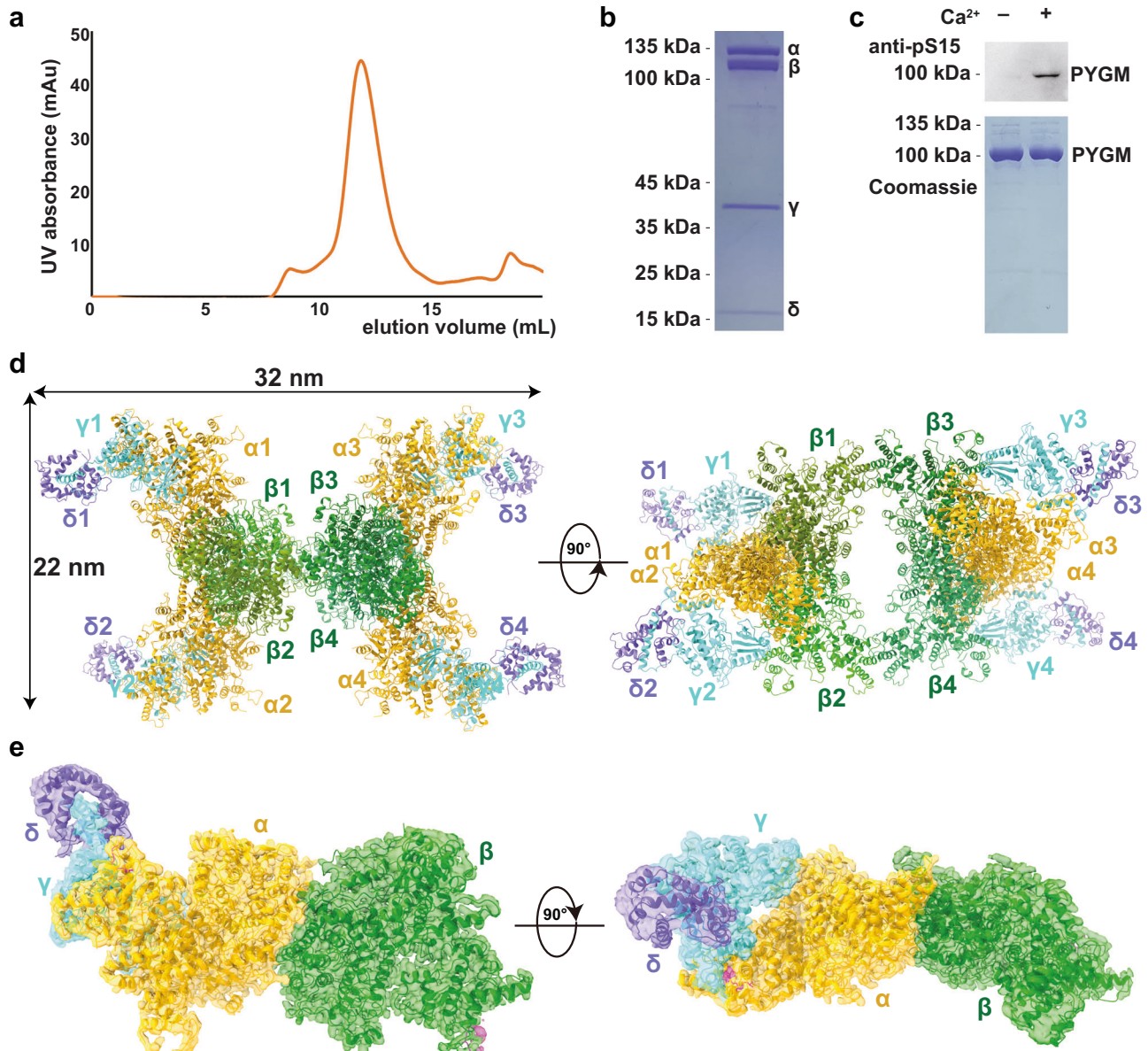

**Fig. 1 | Overall structure of the PhK holoenzyme. a** Size-exclusion chromatography of the inactive PhK holoenzyme. **b** SDS–PAGE analysis of the inactive PhK holoenzyme. This experiment has been repeated at least three times.
**c** Recombinant PhK displays Ca²⁺-dependent kinase activity. The phosphorylation of human muscle phosphorylase (PYGM) was examined by immunoblotting using a phospho-specific antibody. This experiment has been repeated at least three times.

**d** The structural model of the PhK hexadecamer is shown in two orientations. The β-subunits are shown in different shades of green; whereas the α-, γ-, and δ-subunits are shown in gold, cyan, and dark blue, respectively. **e** Local cryo-EM density map of the αβγδ heterotetramer shown in two orientations, with the structural model enclosed.

**Table 1 | Cryo-EM data collection, refinement, and validation statistics**

|  | Inactive_tetramer | Inactive_monomer | Inactive_γδ | Active_tetramer | Active_αγ |
|---|---|---|---|---|---|
| *Data collection and processing* | | | | | |
| Magnification | 81,000 | | | 81,000 | |
| Voltage (kV) | 300 | | | 300 | |
| Electron exposure (e–/Å$^2$) | 60 | | | 60 | |
| Defocus range (μm) | –1.1 to –1.5 | | | –1.1 to –1.5 | |
| Pixel size (Å) | 1.07 | | | 1.07 | |
| Symmetry imposed | C1 | | | C1 | |
| Initial particle images (no.) | 3,769,745 | | | 2,434,165 | |
| Final particle images (no.) | 632,973 | | | 432,047 | |
| Map resolution (Å) | | | | | |
| FSC threshold: 0.143 | 2.92 | 2.72 | 3.12 | 2.94 | 2.91 |
| *Refinement* | | | | | |
| Initial model used (PDB code) | 1CDL | 1CDL | 1CDL | N/A | N/A |
| Model resolution (Å) | | | | | |
| FSC threshold: 0.143 | 2.92 | 2.72 | 3.12 | 2.94 | 2.91 |
| Map sharpening *B* factor (Å$^2$) | 71.3 | 70.7 | 86.2 | 83.9 | 76.6 |
| Model composition | | | | | |
| Non-hydrogen atoms | 81,428 | 20,357 | 5554 | 65,884 | 8090 |
| Protein residues | 10,140 | 2535 | 689 | 8204 | 1018 |
| Ligands | 8 | 2 | 1 | 12 | 1 |
| *B* factors (Å$^2$) | | | | | |
| Protein | 110.94 | 57.96 | 79.66 | 91.36 | 49.51 |
| Ligand | 138.53 | 96.35 | 72.00 | 99.09 | 77.32 |
| R.m.s. deviations | | | | | |
| Bond lengths (Å) | 0.013 | 0.005 | 0.005 | 0.005 | 0.006 |
| Bond angles (°) | 1.214 | 1.046 | 1.081 | 1.006 | 1.039 |
| Validation | | | | | |
| MolProbity score | 1.57 | 1.35 | 1.48 | 1.53 | 1.52 |
| Clashscore | 7.61 | 6.29 | 6.14 | 5.95 | 5.30 |
| Poor rotamers (%) | 0.03 | 0.00 | 0.00 | 0.01 | 0.00 |
| Ramachandran plot | | | | | |
| Favored (%) | 97.16 | 97.97 | 97.22 | 96.78 | 96.39 |
| Allowed (%) | 2.83 | 2.03 | 2.78 | 3.22 | 3.61 |
| Disallowed (%) | 0.01 | 0.00 | 0.00 | 0.00 | 0.00 |

densities, resulting in 2.8 Å and 3.1 Å reconstructions, respectively (Supplementary Fig. 1a). Together, these results enable us to clearly visualize all four subunits of PhK and generate a model for the entire hexadecamer (Fig. 1d). Consistent with previous observations[6,7], the PhK complex exhibits a butterfly-like structure, with a central body consisting of the β$_4$ homotetramer and four wings comprised of the αγδ heterotrimers. The α-subunit interacts with both the β- and γ-subunits to anchor the αγδ subcomplexes onto the main β$_4$ body (Fig. 1e). The δ-subunit or calmodulin is located on the periphery of the PhK holoenzyme and interacts exclusively with the γ-subunit. We also analyzed the cryo-EM structure of active PhK in the presence of Ca$^{2+}$ and determined the structure at a resolution of 2.9 Å (Supplementary Fig. 2, Table 1).

**Structures of the α- and β-subunits**

The structures of the α- and β-subunits are homologous to each other, as suggested by sequence analyses[21]. Each can be further divided into five domains (D1–D5, Fig. 2a–c). The D1 domains display a glucoamylase-like fold[22]. For example, the D1 domain of the α-subunit (D1$_α$) can be superimposed to a glucoamylase-family protein with a root-mean-square deviation (rmsd) of 2.9 Å over 330 residues (Fig. 2d). Furthermore, D1$_α$ appears to possess an intact active site, including the

two Glu (Glu185, Glu371) that participate in catalysis. In contrast, D1$_β$ lacks some of the catalytic residues[23]. However, both the αδ subcomplex (Supplementary Fig. 3a) and the PhK holoenzyme did not show detectable glucoamylase activity in vitro when tested against maltose (a disaccharide of glucose linked by an α–1,4 glycosidic bond), isomaltose (similar to maltose, but with an α–1,6 glycosidic bond), or glycogen (a multibranched polysaccharide of glucose-containing both the α–1,4 and α–1,6 glycosidic bonds) (Fig. 2e). The physiological significance of these glucoamylase-like domains still requires further investigation, especially regarding D1$_α$, as it appears to have a functional active site. On the other hand, it is likely that these domains are responsible for binding to acarbose, a glucoamylase inhibitor, and an anti-diabetic drug that has been demonstrated to bind to PhK and enhance its kinase activity[24].

The D2 domains bear resemblances to a part of a glucanotransferase (Supplementary Fig. 4a), and, therefore may originate from a degenerate glycoside hydrolase. The D3 domains exhibit a helical bundle structure and are the most flexible regions in both the α- and β-subunits. The D4 and D5 domains are homologous to each other (Supplementary Fig. 4b), but display no significant structural similarities to other proteins. The D4 domains play important roles in organizing the intramolecular assembly by interacting with all the

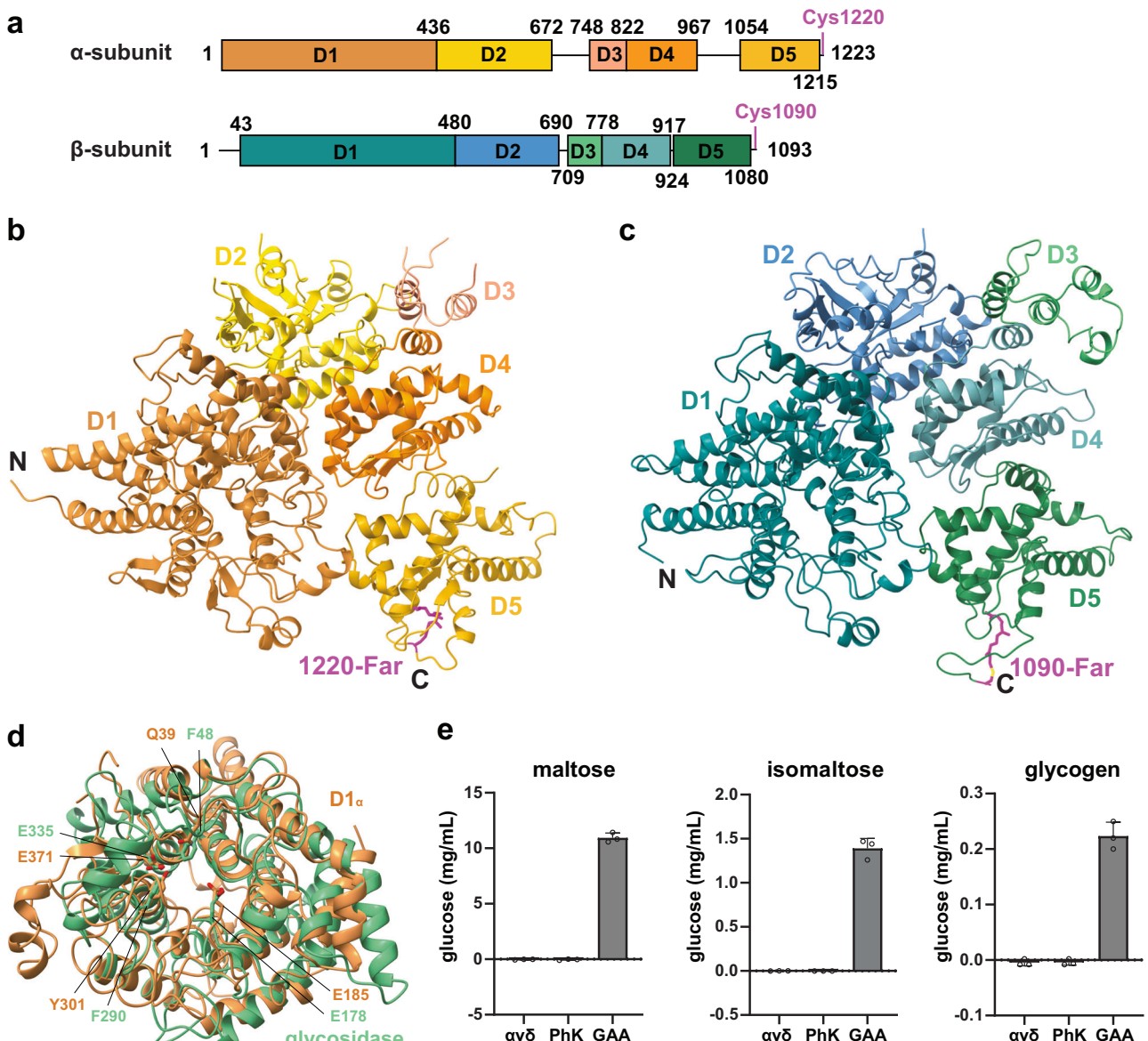

**Fig. 2 | Structures of the α- and β-subunits. a** Schematic representations of the α- and β-subunits. **b** Structure of the α-subunit. The D1–D5 domains, as well as the N- and C-termini, are indicated. **c** Structure of the β-subunit. **d** Structural overlay of the D1 domain of the α-subunit and a glycosidase (PDB ID: 5Z3F), shown in yellow and green, respectively. **e** Neither the αγδ subcomplex nor the PhK holoenzyme has detectable glucoamylase activity against maltose, isomaltose, or glycogen. The human lysosomal α-glucosidase (GAA) is used as a positive control. All data points, as well as the mean ± s.d., are shown. $n = 3$ biological replicates.

other four domains in both the α- and β-subunits (Fig. 2b, c). $D5_\alpha$ is responsible for interacting with the γ-subunit as described below, whereas $D5_\beta$ mediates dimerization of the β-subunits.

## The β$_4$ homotetramer and the α$_2$β$_2$ heterotetramer

A β$_4$ homotetramer lies at the heart of the α$_4$β$_4$γ$_4$δ$_4$ hexadecamer[7,25]. The β$_4$ homotetramer is a dimer of dimers, with each β-subunit interacting with two other molecules (Fig. 3a). The larger dimer interface involving β1/β3 (and the β2/β4 counterpart) buries 1530 Å² surface area in each molecule and involves D5$_\beta$. Both the α- and β-subunits contain a C-terminal CaaX (C, Cys; a, aliphatic residue; X, any residue) prenylation motif, and the corresponding Cys in the α- and β-subunits of rabbit muscle PhK are farnesylated[26,27]. Indeed, mass spectrometry analyses suggest that Cys1220$_\alpha$ and Cys1090$_\beta$ in the human muscle PhK are also farnesylated (Fig. 3b, c). Furthermore, interpretable densities are present for some of these farnesyl groups in the cryo-EM maps (Supplementary Fig. 2g, h). The farnesyl groups on Cys1090$_\beta$ are

buried in the larger dimer interface and glue two D5$_\beta$ domains together (Fig. 3a); whereas the farnesyl group on Cys1220$_\alpha$ is involved in interacting with the γ-subunit as described below. Mutating Cys1090$_\beta$, but not Cys1220$_\alpha$, to Ala leads to a shift of the PhK holoenzyme to a lower molecular weight position on size-exclusion chromatography (Fig. 3d), suggesting that the farnesyl groups on Cys1090$_\beta$ play critical roles in maintaining the structural integrity of the β$_4$ homotetramer and, consequently, the PhK α$_4$β$_4$γ$_4$δ$_4$ hexadecamer.

The smaller β1/β2 (and β3/β4) interface conceals 800 Å² surface area from each subunit and involves regions in D1$_\beta$ and D2$_\beta$. These two β-subunits further interact with two α-subunits, leading to the formation of the α$_2$β$_2$ heterotetramer (Fig. 3e). The α$_2$β$_2$ heterotetramer also features two different types of α/β interactions. The α1/β1 dimer (and also the α2/β2 counterpart) exhibits a pseudo twofold rotation symmetry, and mainly involves residues in D1$_\alpha$ and D1$_\beta$, in particular the Asp68$_\alpha$–Arg94$_\alpha$ and the Arg110$_\beta$–Arg131$_\beta$ helices, burying 1630 Å² surface from each protein. The α1/β2 dimer (and α2/β1) buries 1090 Å²

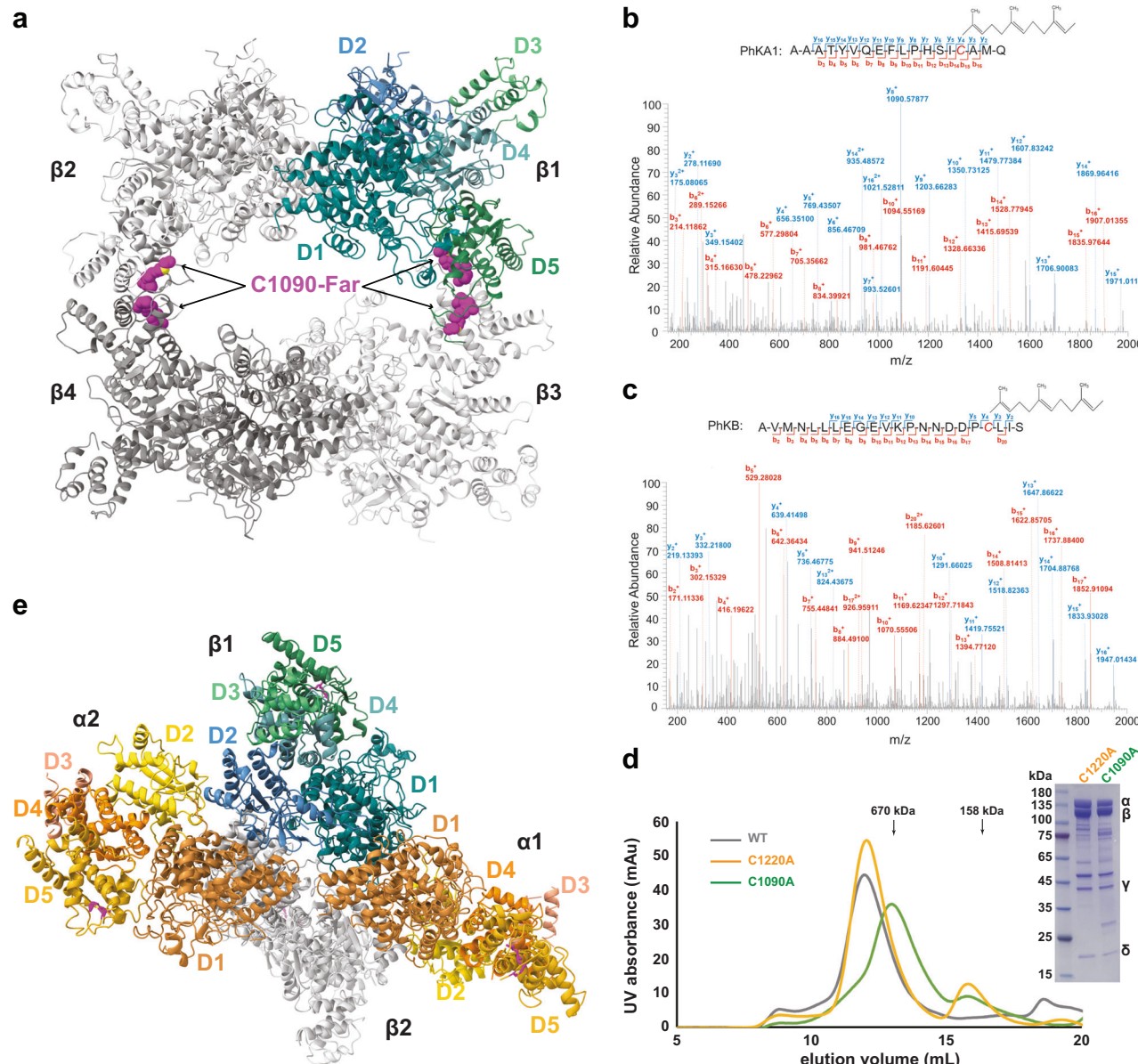

**Fig. 3 | The β₄ homotetramer and the α₂β₂ heterotetramer. a** The β₄ tetramer structure. The five domains of β1 are shown in blue and green, whereas the other three β-subunits are shown in grey. The farnesyl groups are highlighted in magenta. **b** Mass spectrometry analysis suggests that Cys1220 in the α-subunit is farnesylated. **c** Cys1090 in the β-subunit is also farnesylated. **d** Substituting Cys1090β with Ala leads to a shift of the PhK holoenzyme to a lower molecular weight position on size-exclusion chromatography. The elution volumes of two molecular weight standards are indicated. This experiment has been repeated three times. **e** Structure of the α₂β₂ heterotetramer. The five domains of the two α-subunits are shown in yellow and orange. One β-subunit is shown in blue and green, whereas the other is in grey.

surface from each protein, involves the D1α, D2α, and D2β, and is dominated by polar interactions.

**The αγ interaction**

The γ-subunit is attached to the α-subunit; and contrary to previous models, the β- and γ-subunits do not make contact with each other (Figs. 4a, 1e). The γ-subunit features a KD that can be further divided into the N- and C-lobes, and a CRD that contains five helices (αJ−αN, Supplementary Fig. 5a). The KD structure is highly similar to the crystal structure of rabbit PhKγ KD[13,14]. Notably, Leu78γ, Leu90γ, His148γ, and Phe169γ line up and form an intact "regulatory spine" (Supplementary Fig. 5b), demonstrating that the KD adopts an active conformation[28]. The CRD wraps around the kinase C-lobe. The αJ helix interacts with the δ-subunit/calmodulin as described below, whereas the remaining

CRD (referred to as autoinhibitory domain or AID hereafter) loops back and inserts between the C-lobe and α-subunit.

Both KD and AID contribute to interacting with the α-subunit, yielding a large 2590 Å² binding interface between the two proteins. In the KD, N-lobe residues Phe16γ, Tyr17γ, Pro22γ, Arg35γ, Phe99γ, and Phe101γ together accommodate Tyr593α in D2α; whereas Asp52γ forms an ion pair with Lys588α (Fig. 4b). More extensive interactions are present between AID and the α-subunit (Fig. 4c). Two hydrophobic clusters are present on AID: the larger one involves Val328γ, Val333γ, Ile334γ, Leu343γ, Leu346γ, Ile347γ, Tyr350γ, Ala351γ, Ile354γ, Tyr355γ, Trp358γ, Ala369γ, Leu370γ; and the smaller one involves Pro375γ, Ala377γ, and Leu380γ. These hydrophobic residues interact with a range of hydrophobic residues in D5α. The farnesyl group on Cys1220α is also involved in this hydrophobic interface. In addition, >20 salt

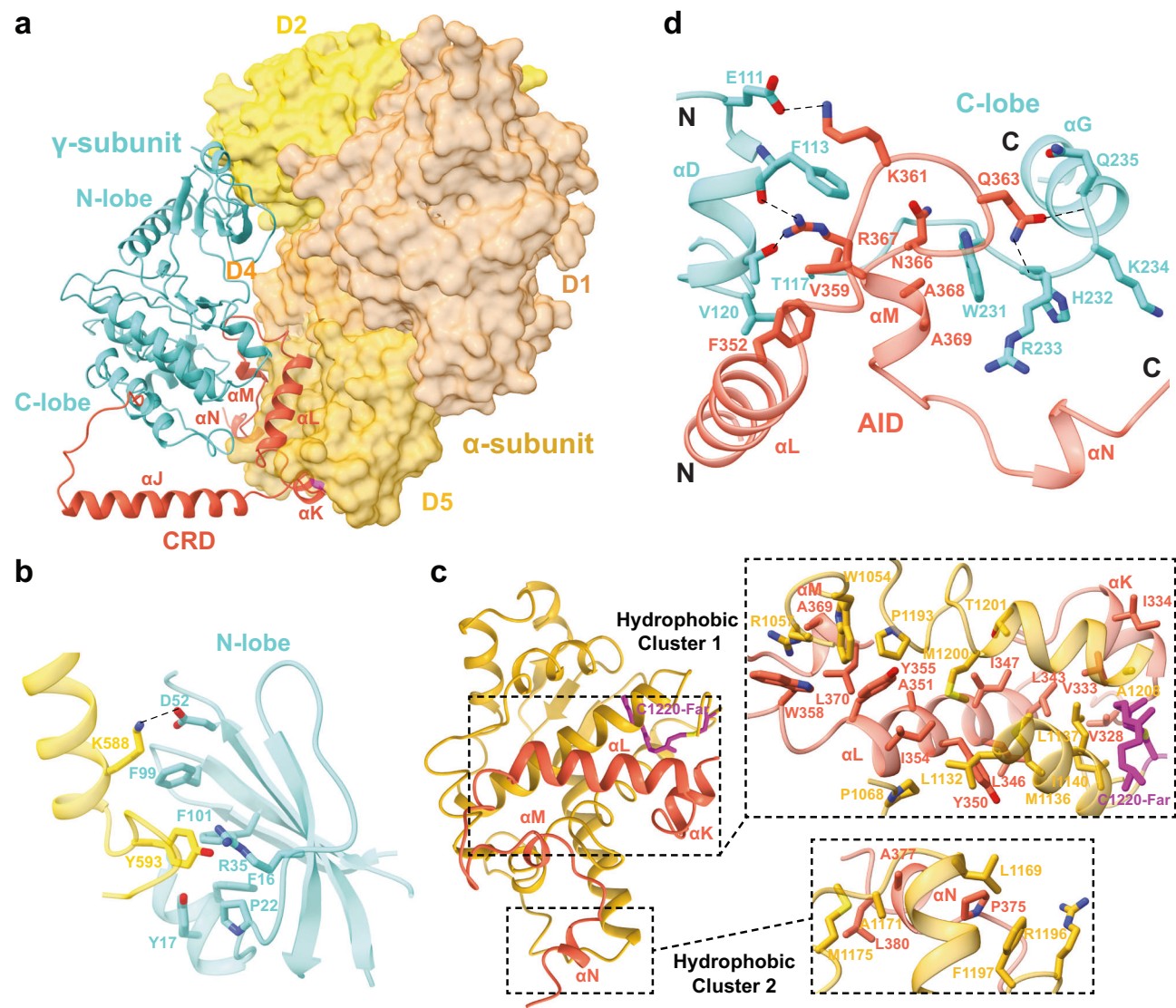

**Fig. 4 | Interactions between the α- and γ-subunits. a** Overall structure of the αγ subcomplex. The α-subunit is shown in a surface representation, whereas the γ-subunit is in ribbons. CRD in the γ-subunit is highlighted in red. **b** Interactions between the kinase N-lobe of the γ-subunit and the α-subunit. Dash lines indicate polar interactions. **c** Interactions between the AID of the γ-subunit and the α-subunit. The farnesyl group on Cys1220α is highlighted in magenta. **d** Interactions between AID and the KD C-lobe.

bridge and hydrogen bond interactions are formed between AID and the α-subunit. Together, these extensive interactions firmly anchor the AID onto the α-subunit.

Importantly, the AID also binds to the kinase C-lobe (Fig. 4d). The αL−αM region interacts with the αD helix, as well as the αF−αG loop. This is highly concordant with a previous mass spectrometry study showing that these regions are not surface exposed[27]. Specifically, Phe352γ and Val359γ cluster with Phe113γ, Thr117γ, and Val120γ. Lys361γ packs against Phe113γ, and also forms a salt bridge with Glu111γ. Gln363γ contacts His232γ−Gln235γ. Asn366γ, Ala368γ, and Ala369γ pack on the Trp231γ, whereas Arg367γ contacts Phe113γ and Thr117γ.

### Autoinhibition of kinase activity

As a result of the interactions described above, Lys361γ−Gly362γ in the αL−αM loop are positioned in such a way that they occupy the −3 and −2 sites of the substrate peptide[14] (Fig. 5a). Thus, the AID would inhibit the KD activity by competitively blocking the binding of the substrate. Notably, this pseudo-substrate mechanism highly resembles the autoinhibition mechanism seen in other Ca²⁺/calmodulin-dependent protein kinases (CAMKs)[29,30].

The truncated γ-subunit lacking the CRD is constitutively active[16,17], demonstrating the autoinhibitory function of this region. To further establish the autoinhibitory role attributed to the AID, we generated a truncation mutant of the γ-subunit consisting of residues 1–326, and prepared its complex with calmodulin (γ₃₂₆δ, Supplementary Fig. 3b). Indeed, γ₃₂₆δ displays Ca²⁺-independent kinase activity towards human muscle GP (PYGM, Fig. 5b). We further designed a triple mutant of the γ-subunit (K361A/Q363A/R367A, 3M) to disrupt the interaction between the αL−αM pseudo-substrate loop and KD (Fig. 4d), and prepared the αγδ subcomplex and PhK holoenzyme accordingly. In contrast to wildtype (WT) αγδ and PhK, which require Ca²⁺ for kinase activity, αγ₃Mδ and PhK₃M exhibit constitutive activity towards PYGM regardless of the presence of Ca²⁺ (Fig. 5b). Together, these data validate our structural analyses, and corroborate the critical roles of AID in autoinhibiting the γ-subunit kinase activity.

### A compact conformation of calmodulin

Calmodulin functions in diverse cellular processes and can versatily bind to hundreds of proteins[31]. A major difference between PhK and other CAMKs is that calmodulin binds to the γ-subunit tightly even in

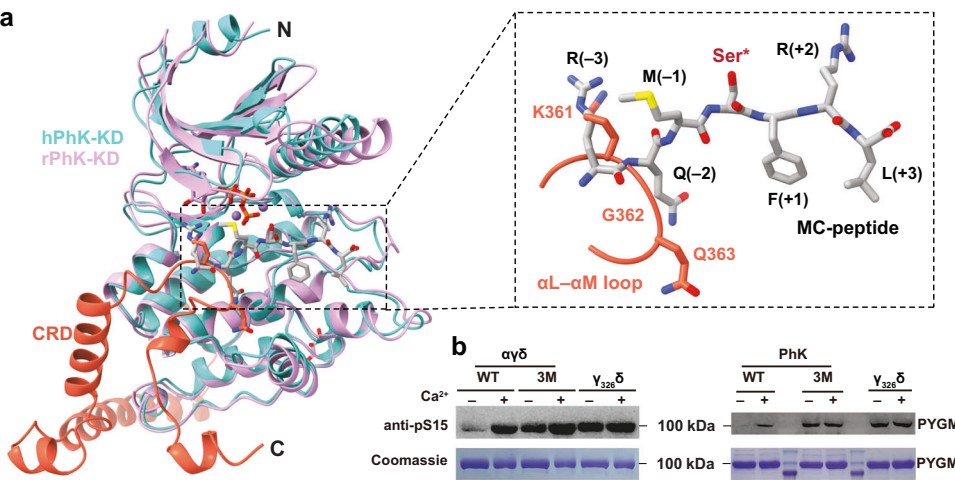

**Fig. 5 | Auto-inhibition of the γ-subunit. a** Structural overlay of human PhKγ and rabbit PhKγ (PDB ID: 2PHK). Lys361$_\gamma$ and Gly362$_\gamma$ occupy the −3 and −2 sites of the substrate peptide (MC-peptide) in rabbit PhKγ. **b** The αγδ subcomplex and PhK holoenzyme containing the K361A/Q363A/R367A (3M) mutations in the γ-subunit, as well as γ$_{326}$δ, all display Ca²⁺-independent kinase activity. This experiment has been repeated three times.

the absence of Ca²⁺ ion[11]. In fact, dissociation between the γ-subunit and calmodulin can only be achieved by protein denaturation[32]. Previous studies suggested two potential calmodulin-binding motifs in the CRD of the rabbit muscle PhK γ-subunit, namely the PhK13 and PhK5 peptides[15], which correspond to residues 303–327 and 343–367 in the human γ-subunit, respectively (Supplementary Fig. 5a). Despite the fact that the PhK5 peptide has been involved in binding Ca²⁺/calmodulin[15,33,34], our structure unambiguously demonstrates that this region is involved in interacting with the α-subunit and KD (Fig. 4a). Instead, the PhK13 peptide, which constitutes the αJ helix, plays an essential role in interacting with calmodulin (Fig. 6a, Supplementary Fig. 5c). This is consistent with previous crosslinking and hydrogen-deuterium exchange studies[10,35].

Calmodulin adopts a compact conformation, and is firmly held between the αJ helix and the kinase C-lobe. It assumes an "antiparallel" wrapping around the αJ helix, with the C- and N-termini of αJ anchored to the N- and C-terminal halves of calmodulin, respectively (Fig. 6b, c). Calmodulin encompasses the majority of the αJ helix. Over half of the αJ residues are hydrophobic and aromatic, allowing for extensive interactions with the inner hydrophobic surface of calmodulin. Notably, calmodulin also engages with the kinase C-lobe (Fig. 6d). Specifically, Lys125$_\gamma$ and Arg128$_\gamma$ in the αE helix of the kinase C-lobe form bonds with Asp57$_\delta$ and Glu68$_\delta$ in the second EF-hand motif of calmodulin. Since these two residues are critical for binding to Ca²⁺, these interactions may offer an explanation for the previous observation that only three Ca²⁺ bind per δ-subunit in PhK[36]. Moreover, Tyr291$_\gamma$, Arg297$_\gamma$, and Phe299$_\gamma$ in the C-lobe–αJ linker contribute to the interaction with calmodulin by interacting with Ala11$_\delta$, Glu12$_\delta$, Lys14$_\delta$, Glu15$_\delta$, Phe17$_\delta$, and Phe66$_\delta$. Together, these extensive interactions effectively tether calmodulin to the γ-subunit.

## Activation of PhK

To investigate the mechanism of Ca²⁺-induced activation of PhK, we analyzed the cryo-EM structure of PhK in the presence of Ca²⁺ (Supplementary Fig. 2, Table 1). When compared to the inactive state structure, the most notable difference is that the KD–αJ region of the γ-subunit and calmodulin were no longer visible (Fig. 7a). Since the protein sample was intact (Supplementary Fig. 3c), these regions likely displayed conformational flexibility and were not discernible after particle averaging. The majority of the AID can still be clearly observed, which remain tightly bound to the α-subunit, except for residues 360–365 that harbor the Lys361$_\gamma$–Gly362$_\gamma$ autoinhibitory sites. Ca²⁺-free calmodulin exists in a compact conformation in the inactive PhK

structure (Fig. 6a). In contrast, a small-angle scattering study suggested that Ca²⁺/calmodulin adopts an extended conformation when bound to the PhK13 peptide[33]. It is conceivable that this Ca²⁺-triggered conformational transition of calmodulin would propagate to the γ-subunit via the tight interaction between these two proteins. The KD was likely detached from the α-subunit–AID platform as a result of this conformational change (Fig. 7b), providing an explanation for its absence in the density map. Importantly, the separation of KD from AID would lead to kinase de-inhibition and activation.

To investigate this scenario biochemically, we introduced a tobacco etch virus (TEV) protease cleavage site between the αJ helix and AID in the γ-subunit (Fig. 7c, Supplementary Fig. 5a) and generated the αγ$_{TEV}$δ complex. A twin-strep tag on the N-terminus of the α-subunit was utilized for a pull-down experiment. Our rationale was that in the absence of Ca²⁺, the KD would be pulled down by the α-subunit even after separation from AID by the TEV protease, owing to its extensive interactions with AID and the α-subunit (Fig. 4b, d). Conversely, significantly less KD should remain associated after activation, due to its detachment as described above. This hypothesis was confirmed by the experiment. As shown in Fig. 7c, prior to TEV cleavage, the intact γ-subunit and calmodulin were tightly associated with the α-subunit (left panel), irrespective of the presence of ethylenediamine-tetraacetic acid (EDTA) or Ca²⁺. Even after the separation of KD from AID by TEV protease, KD remained bound to the α-subunit in the presence of EDTA (right panel). However, in the presence of Ca²⁺, KD was not pulled down by the α-subunit after TEV cleavage, and only AID remained associated with the α-subunit. Calmodulin appeared to be less tightly associated with the KD–αJ module after TEV cleavage in the presence of EDTA. This observation may be attributed to the combination of EDTA and the disrupted end of αJ, as calmodulin consistently exhibited tight binding to the intact γ-subunit even in the presence of EDTA (Fig. 7c, left panel), and the γ$_{326}$ truncation mutant in the presence of Ca²⁺, as evidenced in the γ$_{326}$δ sample (Supplementary Fig. 3b).

Taken together, our structural and biochemical data suggest that calmodulin likely regulates PhK activity using a spring-loaded mechanism: Ca²⁺-free calmodulin binds to the γ-subunit in a compact conformation, resembling a compressed spring; upon sensing Ca²⁺, calmodulin undergoes a conformational change and facilitates detachment of KD, thereby de-inhibiting kinase activity (Fig. 7b).

ADP has also been shown to allosterically enhance PhK activity by binding to the β-subunit[37]. Indeed, the active PhK structure clearly reveals the presence of ADP bound to the β-subunits, which can be

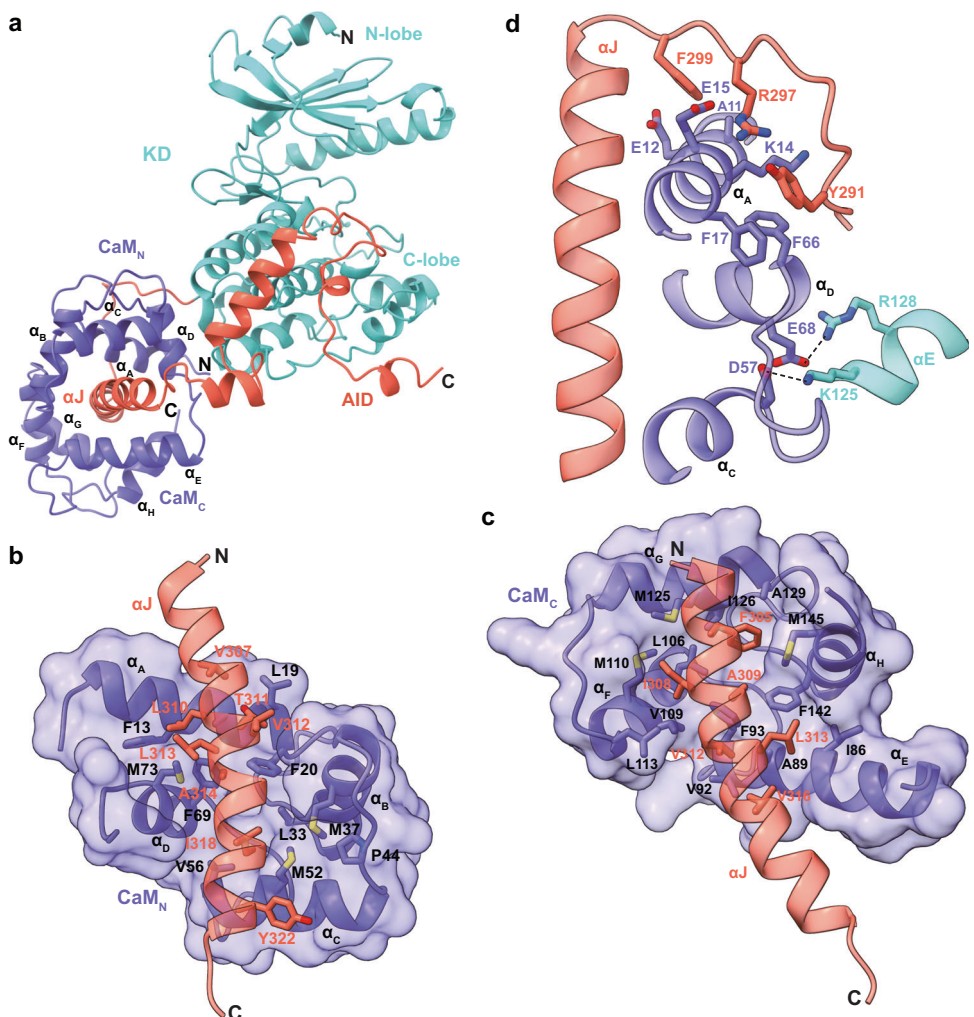

**Fig. 6 | Structure of the γδ subcomplex. a** Overall structure of the γδ subcomplex shown in ribbons. The eight helices in calmodulin are labeled $\alpha_A$–$\alpha_H$. **b** Interactions between αJ and the N-terminal half of calmodulin. **c** Interactions between αJ and the C-terminal half of calmodulin. **d** Interactions between the kinase C-lobe and calmodulin.

attributed to the ADP present in the sample buffer. Within the structure, ADP is nestled between the D1β, D2β, and D4β domains (Fig. 7d). The diphosphate moiety is surrounded by Arg211β, Tyr213β, Arg214β, and Arg274β, which effectively neutralize the negative charges of the phosphate groups and allow the binding of ADP without divalent metal ions. The presence of ADP leads to a slight conformational change of D2β–D5β relative to D1β (Fig. 7e). Notably, D2β interacts with D2α to establish the α1/β2 and α2/β1 dimers (Fig. 3e); whereas D2α interacts with the KD N-lobe and contributes to the maintenance of the inactive state of the γ-subunit (Fig. 4a). The ADP-induced conformational changes in the β-subunits would propagate to the α-subunits through the D2β–D2α interaction, facilitating the disruption of the α-subunit–KD interaction and thereby allosterically enhancing PhK activity (Supplementary Movie 1).

## Discussion

The investigation into glycogen metabolism and glycogen phosphorylase[38–40] has led to the discovery of phosphorylase kinase nearly 70 years ago[41], positioning it at the crossroads of energy supplementation and muscle contraction, as well as at the intersection of signaling and metabolic pathways[42]. In this study, we aimed to elucidate the high-resolution structures of human muscle phosphorylase kinase in both its inactive and Ca²⁺-active conformations. Our findings reveal that the complex is comprised of a tetramer of the αβγδ

tetramer. Notably, the architectural details of the large α- and β-subunits clearly reveal that the D1 domains of the α- and β-subunits exhibit glucoamylase-like folds. Intriguingly, despite the presence of catalytic residues in D1α, we did not detect hydrolase activity. We have also demonstrated the presence of farnesylation on the CaaX motifs of the α- and β-subunits. Farnesyl groups are known to facilitate the membrane attachment of various proteins, including the Ras family of GTPases[43]. However, in our structure, the farnesyl groups are embedded within protein-protein interaction interfaces, raising questions about their potential exposure and their role in regulating PhK localization. Further studies are required to examine these aspects.

Our results also shed light on the autoinhibition and activation mechanisms of PhK, which bear a resemblance to the strategies observed in other Ca²⁺/calmodulin-dependent protein kinases (CAMKs)[29]. For instance, in CaMKI[44] or CaMKII[45], a regulatory segment that obstructs the active site binds to the C-lobe of the kinase domain, akin to the AID in the PhK γ-subunit. Kinase activation is achieved through the binding of Ca²⁺-loaded calmodulin, resulting in the separation of the kinase domain and the regulatory segment. However, a notable difference is that in PhK, calmodulin remains associated with PhK even in the absence of Ca²⁺. In our study, we observed that the Ca²⁺-free calmodulin adopts a compacted conformation and establishes extensive contacts with the KD–αJ module in the γ-subunit. Upon binding to Ca²⁺, calmodulin is presumed to undergo a

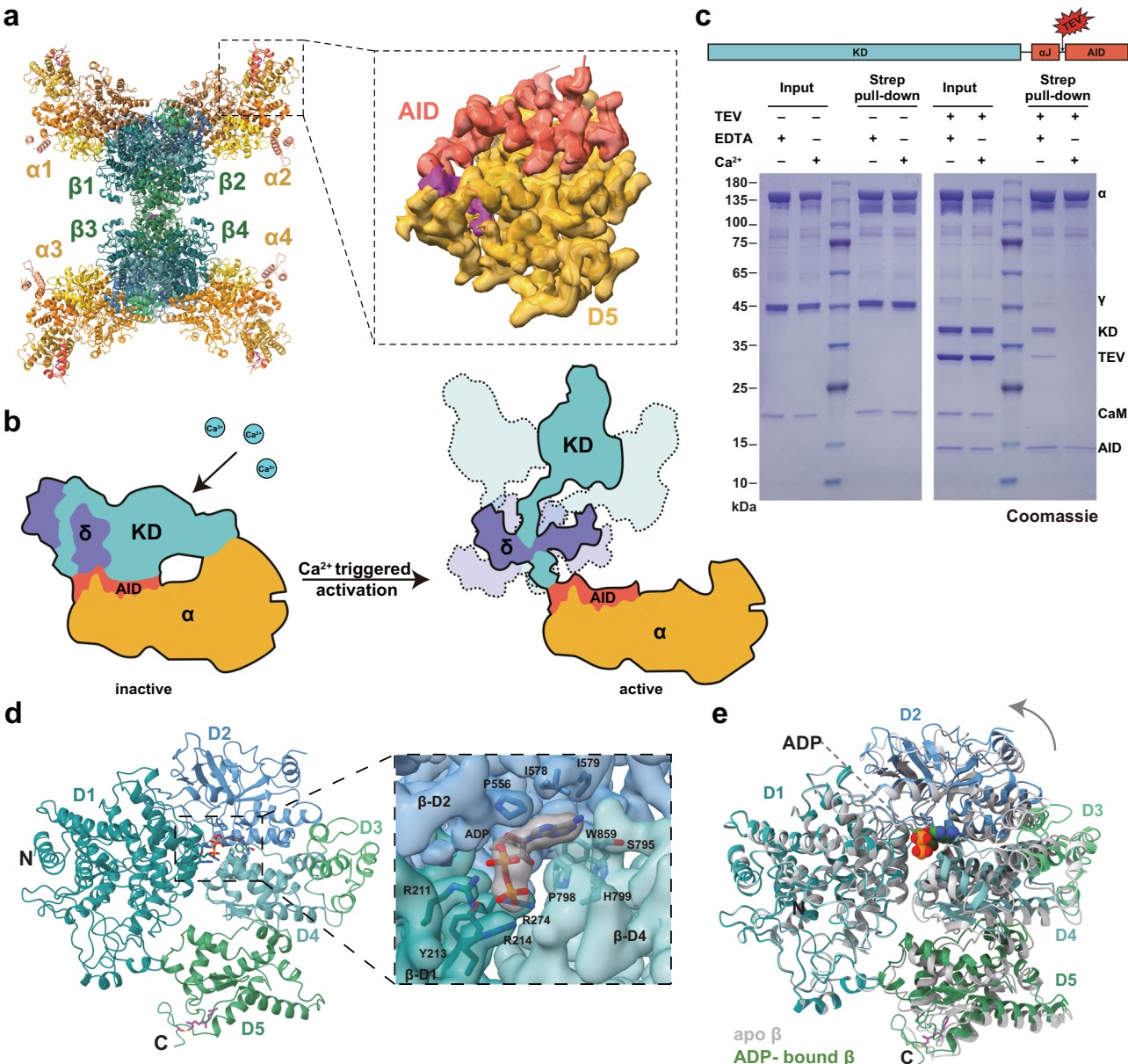

**Fig. 7 | Insights into PhK activation. a** Cryo-EM structure of PhK in the presence of Ca²⁺. The density map of the D5ₐ−AID region is shown on the right. **b** A model of Ca²⁺-triggered de-inhibition and activation of PhK. Ca²⁺-free calmodulin binds to the γ-subunit in a compact conformation. Upon sensing Ca²⁺, calmodulin undergoes a conformational change and facilitates detachment and de-inhibition of KD. **c** A TEV protease cleavage site was introduced into the γ-subunit between αJ and AID. After

TEV cleavage, KD was not pulled down by the α-subunit in the presence of Ca²⁺. This experiment has been repeated at least three times. **d** Overall structure of the ADP-bound β-subunit. The cryo-EM density map of the ADP-binding pocket is shown on the right. **e** Structural overlay of the apo (from the inactive PhK structure) and ADP-bound β-subunits.

conformational change, leading to the release of the autoinhibition of the γ subunit (Fig. 7b).

Calmodulin is known for its ability to adopt various conformations when interacting with a diverse range of proteins[46,47]. In the structures of calmodulin binding to peptides such as CaMKII[48] or MLCK[49], Ca²⁺-bound calmodulin adopts a compact conformation, wrapping around an α-helix, similar to what is observed in our Ca²⁺-free PhK structure. However, due to the flexibility of the central linker between its N- and C-terminal halves, calmodulin can also exhibit extended conformations, even in the presence of Ca²⁺. For instance, eEF-2K is another kinase that relies on calmodulin for activation, and the Ca²⁺/calmodulin−eEF-2K structure reveals that calmodulin displays an extended structure and primarily binds to eEF-2K via its C-terminal half[50,51]. A prior small-angle X-ray scattering study has demonstrated

that Ca²⁺/calmodulin adopts an extended conformation when binding to the αJ helix of PhKγ[33]. Further investigation is required to fully understand the specific conformation of calmodulin in this context and how it contributes to the activation of PhK.

In addition to Ca²⁺ ions, ADP also functions as an allosteric enhancer of PhK activity, and uniquely targets the β-subunit[37]. The allosteric ADP site has now been elucidated, revealing that ADP is surrounded by several positively charged residues in the β-subunit, independently of divalent metal ions. Notably, a recent study has demonstrated that ADP serves as an allosteric activator for eEF-2K, and is similarly located in a pocket surrounded by basic residues[51]. Our previous research has demonstrated that ATP acts as an allosteric regulator of the pseudokinase Fam20A[52]. The potential multifunctional roles of ADP/ATP as small molecule allosteric regulators, beyond their

canonical roles as energy and phosphate donors, also warrant further investigation in future studies.

PhK can be activated through other mechanisms, including phosphorylation. In fact, the connection of PhK with epinephrine signaling represents an archetype of signaling cascade[53,54]. Both the α- and β-subunits are phosphorylated by PKA, as well as PhK itself[8,55]. However, the specific contributions of these modifications to the regulation of the γ-subunit remain poorly understood. Furthermore, studies have reported that the α- and β-subunits are dephosphorylated by different phosphatases[56,57], suggesting the inactivation of PhK may also be intricately regulated. Given the emerging functional importance of glycogen in various physiological processes, including adipocyte homeostasis[58], macrophage function[59], liver tumor initiation[60], it is likely that much more will be learned regarding the physiology and regulation of PhK.

In summary, our work has provided a structural framework for a mechanistic understanding of PhK, revealing that the autoinhibition of the γ-subunit kinase activity is achieved through a pseudo-substrate mechanism. We have also shown that Ca$^{2+}$-induced PhK activation is likely achieved through the release of AID-mediated KD autoinhibition, as a result of a Ca$^{2+}$-dependent conformational change of calmodulin. These findings not only address long-standing questions in PhK biochemistry but also establish a foundation for future studies on this important kinase complex.

## Methods

### Plasmids

To generate the PhK holocomplex, the DNA fragments encoding full-length PhK α-, β-, γ-subunits and calmodulin were cloned into pMLink vector (a kind gift from Professor N. Gao, Peking University) for expression in HEK293F cells. A twin-strep tag was engineered to the N-terminal region of the β-subunit. These DNA fragments were then linked into one plasmid following a previously described procedure[61]. Briefly, 1 μg of one plasmid was digested overnight with 10 units PacI (New England Biolabs) in 20 μL at 37 °C, while the other was digested at 25 °C with 10 units SwaI (New England Biolabs). Enzymes were inactivated at 65 °C for 20 min. The PacI digest was treated with 1 unit T4 DNA polymerase (New England Biolabs) and 2 mM dCTP, and the SwaI digest was treated with 1 unit T4 DNA polymerase and 2 mM dGTP, respectively, both at 12 °C for 15 min. EDTA was added to these mixtures to a final concentration of 10 mM before heat inactivation was performed at 75 °C for 20 min. Finally, the two plasmids were mixed and heated to 65 °C and cooled to room temperature for annealing, followed by E. coli transformation. Mutations were introduced into the expression plasmids by a PCR-based method. For the αγδ subcomplex, a twin-strep tag was engineered to the N-terminal region of the γ-subunit.

To generate the γ$_{326}$δ complex, the DNA fragment encoding residues 1–326 of the γ-subunit was cloned into a modified pQLink vector with an N-terminal GST tag and TEV protease cleavage site for expression in E. coli. The DNA fragment encoding full-length calmodulin was cloned into a modified pQLink vector with an N-terminal 8×His tag and TEV cleavage site. These two genes were linked into one plasmid following the procedure described above.

The DNA fragment encoding full-length human muscle phosphorylase (PYGM) was cloned into a modified pCS2 vector with an N-terminal twin-strep tag and TEV cleavage site. The DNA fragment encoding human lysosomal α-glucosidase (GAA) residue 70-952 was cloned into the pMLink vector, and engineered with an N-terminal IL-2 signal peptide and 8×His tag. The primers used in this study were listed in Supplementary Data.

### Protein expression and purification in HEK293F cells

HEK293F cells were cultured in SMM 293T-I medium (Sino Biological Inc.) at 37 °C, with 5% CO$_2$ and 55% humidity. The plasmid expressing the PhK complex was transfected into the cells using polyethylenimine (PEI, Polysciences). After 36 hours of transfection, the cells were collected by centrifugation at 500 × $g$ and resuspended in buffer A [50 mM HEPES, pH 6.8, 100 mM NaCl, 30 mM β-glycerolphosphate, 10% w/v sucrose and 2 mM dithiothreitol (DTT)], supplemented with Protease Inhibitor Cocktail (Bimake). The cells were then disrupted by sonication. The insoluble debris was removed by centrifugation, and the resulting supernatant was incubated with the StrepTactin beads (Smart Lifesciences) at 4 °C for 1 h with gentle shaking. The beads were washed with buffer A supplemented with 10 mM EDTA and eluted using buffer A supplemented with 5 mM desthiobiotin (IBA). Next, the purified protein was treated with the λ-phosphatase in the presence of 2 mM MnCl$_2$ at room temperature for 1 hour. The protein was incubated with extra purified calmodulin and further purified by gel filtration chromatography using a Superose 6 Increase 10/300 GL column (GE Healthcare). The PhK mutant and the αγδ subcomplexes were purified similarly.

For the purification of PYGM, the cell lysate was incubated with the StrepTactin beads in buffer B (50 mM Tris-HCl, pH 8.0, 500 mM NaCl, 10 % v/v glycerol, and 2 mM DTT), supplemented with 1 mM phenylmethylsulfonyl fluoride (PMSF). The beads were washed with buffer B, and the elution was performed using buffer B supplemented with 5 mM desthiobiotin. The eluted protein was digested with the TEV protease overnight and further purified by gel filtration chromatography using a Superose 6 Increase 10/300 GL column, eluted in buffer C (25 mM HEPES, pH 7.5, 150 mM NaCl and 2 mM DTT).

For the purification of GAA, the plasmid was transfected into the HEK293F cells using PEI. The conditioned medium was collected 4 days after transfection, buffer-exchanged into buffer D (25 mM Tris-HCl, pH 7.4, 150 mM NaCl), and then incubated with Ni-NTA resin at 4 °C for 1 h. The beads were subsequently washed with buffer D supplemented with 20 mM imidazole, and eluted by buffer D supplemented with 300 mM imidazole. The eluted protein was purified by gel filtration chromatography using a Superdex 200 Increase 10/300 GL column (GE Healthcare), eluted in buffer D.

### Protein expression and purification in *E. coli*

The *E. coli* BL21(DE3) bacterial cultures were grown at 37 °C in the LB (Luria-Bertani) medium until reaching an OD$_{600}$ of 0.6–0.8 before induced with 0.5 mM isopropyl β-D-1-thiogalactopyranoside (IPTG) at 18 °C for 20 h. The cells were collected by centrifugation at 5000 × $g$ and were resuspended in buffer E (50 mM Tris-HCl, pH 8.0, 500 mM NaCl, and 5 mM β-mercaptoethanol). The cells were then disrupted by sonication in the presence of 1 mM PMSF. Following sonication, the insoluble debris was removed by centrifugation. For the purification of the γ$_{326}$δ complex, the supernatant was incubated with the glutathione beads (DiNing) at 4 °C for 1 hour with gentle shaking in the presence of 0.5 mM CaCl$_2$, and the recombinant protein was eluted using buffer E supplemented with 10 mM reduced L-glutathione (Sigma–Aldrich). The purified protein was digested with TEV protease overnight. The untagged protein was further purified by gel filtration chromatography using a Superdex 200 Increase 10/300 GL column, eluted in buffer F (25 mM HEPES, pH 6.8, 150 mM NaCl, and 2 mM DTT). The purified protein was then incubated with glutathione beads to remove the GST fusion tag, followed by another gel filtration chromatography step and eluted using buffer F. For the purification of calmodulin, the cell lysates in buffer E were incubated with Ni-NTA resin (GE Healthcare), washed using buffer E supplemented with 20 mM imidazole, and eluted using buffer E supplemented with 300 mM imidazole. The purified calmodulin was digested with TEV protease, and the untagged calmodulin was further purified by gel filtration chromatography using a Superdex 200 Increase 10/300 GL column, eluted in buffer C.

## Cryo-electron microscopy sample preparation

For preparing the cryo-EM sample of inactive PhK, dephosphorylated PhK was incubated with purified calmodulin using 1:20 molar ratio in the presence of 4 mM EGTA, and further purified using a Superose 6 Increase 10/300 GL column, eluted in buffer F. Before cryo-EM sample preparation, 4 mM EGTA and additional calmodulin (fourfold excess) were added to the protein sample, to ensure that calmodulin does not fall off. For the sample of $Ca^{2+}$-activated PhK, the dephosphorylated protein was first incubated with excessive calmodulin as described above, in the presence of 1 mM $CaCl_2$ on ice for 1 hour, and then purified by Superose 6 Increase 10/300 GL column, eluted in buffer G (25 mM HEPES, pH 8.2, 150 mM NaCl, 1 mM $CaCl_2$, and 2 mM DTT). Before cryo-EM sample preparation, additional calmodulin (fourfold excess) was added, and the mixture was incubated on ice for 1 hour, in the presence of 1 mM $CaCl_2$, 1 mM ADP, 1 mM $AlCl_3$, 4.5 mM NaF, 2 mM $MgCl_2$.

Holey carbon grids (Quantifoil R1.2/1.3) were coated by continuous carbon film using a mica plate and glow-discharged for 30 seconds using a plasma cleaner (Harrick PDC-32G-2). The sample preparation was carried out using a Vitrobot Mark IV (FEI). Specifically, 4 μL aliquots of the PhK holoenzyme (0.45 mg/ml) were applied onto the grids, allowing for a 5-second incubation at 4 °C and 100% humidity. Subsequently, the grids were blotted with filter paper (Tedpella) for 0.5 seconds using a blotting force of −1, and immediately plunged into liquid ethane. Grid screening was performed using a 200 kV Talos Arctica microscope equipped with Ceta camera (FEI). Good grids were transferred to a 300 kV Titan Krios electron microscope (FEI) for data collection.

Data were acquired using the EPU software (E Pluribus Unum, Thermo Fisher) on a K3 Summit direct electron detector (Gatan) operating in a super-resolution mode. The defocus range was set from −1 to −1.5 μm. Micrographs were recorded at a nominal magnification of 81,000 (pixel size of 1.07 Å at the object scale). The micrographs were dose-fractioned into 40 frames with dose rate of ~21.47 electrons per pixel per second for a total exposure time of 3.2 s.

## Imaging processing

The data were processed using cryoSPARC (version 3.2.1)[62]. Movie stacks were collected and motion-corrected using the patch motion correction, and the CTF parameters were determined using the patch CTF estimation. Summed images were then manually screened to remove low-quality images through exposure curation. Initially, particle picking was performed using a blob picker, and templates were generated through subsequent 2D classification. The particles from template picking were subjected to multiple rounds of 2D classification to exclude inaccurate particles. The selected particles were then subjected to ab initio reconstruction and heterogeneous refinement to select the appropriate particles. The particles resulting from the heterogeneous refinement were then utilized in homogeneous refinement, ultimately leading to the generation of the final 3D reconstruction. To obtain a higher-quality local density map, mask-based local refinement was further performed using cryoSPARC. All maps were then sharpened using DeepEMhancer[63]. The local resolution map was produced with the local resolution estimation program in cryoSPARC and displayed using UCSF Chimera[64].

## Model building and structure refinement

Initial models of PhK α- and β-subunits and the KD of γ subunits were generated using AlphaFold2[65,66], and calmodulin was obtained from a previous crystal structure (PDB ID: 1CDL)[49]. These models were docked into the cryo-EM density map using UCSF Chimera. The CRD of the γ-subunit was built de novo using Coot[67]. Further structure model building was performed using Coot and refined using the real-space refinement in Phenix[68]. Figures were prepared with UCSF ChimeraX[69].

## Mass spectrometry

PhK was denatured using 8 M Urea, treated with DTT, and alkylated with iodoacetamide in the dark, followed by overnight trypsin digestion. After desalting, the sample was dried and analyzed by LC-MS/MS using the Orbitrap Fusion™ Lumos™ Tribrid™ mass spectrometer (Thermo Fisher Scientific). MS/MS spectra were searched using Proteome Discoverer software (version 2.1) with a UniportKB human database. The precursor mass tolerance was set at 15 ppm, and a false discovery rate (FDR) of 1% was applied at both the peptide and protein levels. Trypsin was specified as the protease, and up to two missed cleavages were allowed. The mass spectrometry data are provided in Supplementary Data.

## Kinase assay

A concentration of 100 nM of PhK, $PhK_{3M}$, or $γ_{326}δ$, or 40 nM of αγδ, $αγ_{3M}δ$, and $γ_{326}δ$ subcomplex were incubated with 2 μM PYGM in a reaction buffer containing 25 mM HEPES, 100 mM NaCl, 1 mM ATP, 5 mM $MgCl_2$, and either 0.5 mM EGTA at pH 6.8 or 1 mM $CaCl_2$ at pH 8.2. The assays were carried out at 30 °C for 10 min and terminated by the addition of SDS-PAGE sample buffer (TransGen Biotech) supplemented with 20 mM EDTA, and then boiled. The samples were then resolved by SDS-PAGE and analyzed by immunoblotting using the anti-PYGL (phospho S15) antibody (Abcam, ab227043, 1:1000).

## Glucoamylase assay

To determine glucoamylase activity, 100 nM PhK holoenzyme or 400 nM αγδ subcomplex was incubated with 15 mg/mL maltose (Macklin), isomaltose (Macklin) or glycogen (from bovine liver, Sigma−Aldrich) at 37 °C for 2 hour in the reaction buffer containing 50 mM HEPES, pH 6.8, 100 mM NaCl. The activity was measured by quantifying the amount of released glucose using the Glucose Assay Kit (Sigma−Aldrich). Briefly, the concentration of generated glucose was diluted to <0.1 mg/mL using the reaction buffer. The diluted samples were reacted with the Assay Reagent at 37 °C for 30 min and subsequently mixed with sulfuric acid. The absorbance was measured at 540 nm using the BioTek Cytation 5 (Agilent) in 96-well plate. GAA (400 nM) was used as a positive control, and the reaction buffer for GAA contained 50 mM sodium acetate, pH 4.5, and 100 mM NaCl.

## TEV cleavage and pull-down assay

A TEV protease cleavage site (GENLYFQGG) was introduced into the γ-subunit between Lys325 and Pro326. Additionally, the γ-subunit was modified with an N-terminal 6×His tag and a C-terminal Flag tag. This TEV-engineered αγδ complex ($αγ_{TEV}δ$) with a twin-strep tag on the α-subunit was purified using the above-described procedure. For the pull-down experiments, the $αγ_{TEV}δ$ complex (0.03 mg/mL) was initially incubated with or without TEV protease (in a threefold excess molar ratio), and with 0.2 mM EDTA or with 1 mM $CaCl_2$ at 30 °C for 1 hour. Subsequently, the treated samples were incubated with StrepTactin beads at 4 °C for 1 hour with gentle shaking in the binding buffers (25 mM HEPES, pH 6.8, 150 mM NaCl, 2 mM DTT), supplemented with 10% glycerol and with either 0.2 mM EDTA or 1 mM $CaCl_2$. The beads were then centrifuged at $500 × g$ and washed three times with the corresponding binding buffers. The proteins bound to the beads were eluted using the binding buffer supplemented with 10 mM desthiobiotin. The eluted proteins were subsequently analyzed by SDS-PAGE and visualized through Coomassie Blue staining.

## Reporting summary

Further information on research design is available in the Nature Portfolio Reporting Summary linked to this article.

# Data availability

The data that support this study are available from the corresponding authors upon request. Cryo-EM density maps of PhK have been

deposited in the Electron Microscopy Data Bank with accession codes EMD-36212 (inactive, overall holoenzyme), EMD-36214 (inactive, αβγδ subcomplex), EMD-36215 (inactive, γδ subcomplex), EMD-36213 (Ca²⁺, overall holoenzyme), and EMD-36216 (Ca²⁺, αγ subcomplex). Structural coordinates have been deposited in the Protein Data Bank with the accession codes 8JFK (inactive, holoenzyme), 8XYA (inactive, αβγδ subcomplex), 8XYB (inactive, γδ subcomplex), 8JFL (Ca²⁺, holoenzyme) and 8XY7 (Ca²⁺, αγ subcomplex). Uncropped gels and blots underlying Figs. 1b, c, 2, 3f, and Supplementary Fig. 3b are provided in the Source Data file. Source data are provided in this paper.

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

## Acknowledgements

We are grateful to the Core Facilities at the School of Life Sciences, Peking University, for their assistance with negative-staining EM; the Cryo-EM Platform of Peking University for their support with data collection; and the High-performance Computing Platform of Peking University for their aid with computation. We also acknowledge the National Center for Protein Sciences at Peking University for their help with the BioTek Cytation Reader and mass spectrometry facilities. This work received support from the National Natural Science Foundation of China (32325018) and the Qidong-SLS Innovation Fund to J.X., as well as from Changping Laboratory.

## Author contributions

X.Y. and M.Z. contributed equally to this work. X.Y. and M.Z. carried out the structural and biochemical studies, with Y.W. providing critical assistance. X.L. performed mass spectrometry analyses. J.X. conceived and supervised the project. J.X., X.Y., and M.Z. wrote the manuscript, with inputs from all authors.

## Competing interests

The authors declare no competing interests.
