## [Peer Review File · Nature Communications]

Architecture and activation of human muscle phosphorylase kinaseReviewer #1 (Remarks to the Author):

This is an amazing manuscript that completes a story that began over 70 years ago. At the same time it also opens the door for a new chapter in understanding the complex mechanism that controls the activity of this essential protein kinase that is so fundamental for metabolism. The manuscript is clearly written and the cryoEM results are solid. I have no technical concerns whatsoever with this manuscript. The writing is actually exemplary. It is obviously a very complex macromolecular complex that describes many levels of regulation, but the authors have distilled down the essential features in a very concise way. The regulation by calmodulin is especially intriguing and surprising. The evolutionary insights into the alpha subunit are also intriguing. Is there a missing substrate that is hydrolyzed? The allosteric ADP site is another intriguing discovery. Is this a novel ADP/ATP binding site? It is extremely unusual to have the phosphates buried in so many basic residues.

While phosphorylase kinase was the first kinase to be discovered by Krebs and Fischer in the 1950s, their work was built on the remarkable foundation that was laid by Gerti Cory and Arda Green in the 1940s. Louise Johnson was also one of my close personal friends as well as Gerry Carlson, so I have followed this structure very closely for many decades and appreciate fully how challenging it has been. Obviously one cannot understand any kinase fully until you have the complete complex with all the domains, and this structure shows why the earlier studies were incomplete. This cryoEM structure has been long awaited and intriguing because a number of different states have been captured. I will delve more deeply into the activation mechanism associated with kinase activation, but the model proposed seems quite reasonable.

This paper should be published quickly and highlighted as an historical landmark!

Dr. Susan Taylor

Reviewer #2 (Remarks to the Author):

This is a well written article that expands our knowledge on the sub-unit organisation and structure of phosphorylase kinase. The work is a nice extension of previous studies on the crystal and EM structures of this enzyme. While the new data is compelling and well conducted, I am afraid that I consider this to be an incremental advance in the field.

Reviewer #3 (Remarks to the Author):

In this manuscript, Yang, Zhu et al. report the structure of the human muscle phosphorylase kinase complex. The complex is formed from a dimer of dimers in which the beta subunits form the core and the gamma and delta (kinase and calmodulin, respectively) subunits on the periphery. The authors describe key observed interactions and protein-protein interaction interfaces. Ca-free CaM molecules are bound to the C-tail of the gamma subunits in a region that is near an autoinhibitory domain. Upon addition of Ca²⁺, the KD and CaM are no longer observed in cryo-EM reconstructions, suggesting that separation of the KD from the autoinhibitory domain by CaM upon binding Ca²⁺ is the mechanism for kinase activation. While limited on biochemistry, this is a high-quality structural paper with clear figures for a physiologically important enzyme. However, I feel the proposed spring-loaded mechanism of activation needs additional validation. It is also difficult to understand the significance and what advancement to the field this research offers without a dedicated discussion section.

1) Are the farnesyl groups needed for formation of the complex? What effect does mutation of the Cys in the CaaX box have?

2) Regarding activation of PhK by Ca²⁺, it is stated that the "protein sample was intact" but references a prior size exclusion run (Fig. 1b) that is not in the presence of Ca²⁺. How are you sure that CaM is still attached but is not observed due to flexibility? I don't think the Superose6

chromatogram for that sample is shown. My concern is that dissociation of CaM could yield the same result (no observed density for CaM or the KD), so you have to provide evidence that the complex is still intact.

3) I do not think that the cleavage experiment shown in Fig. 5C is strong evidence of the mechanism you are proposing. A cartoon in the main text may help to visualize your construct rather than just a supplemental sequence alignment. I think you have created KD-TEV site-CRD-AID-FLAG +CaM +alpha complex. Your assumption is that with Ca²⁺, the kinase domain will be released if the TEV site is cleaved. This seems reasonable based on your Ca²⁺ structure, but this structure also does not have a resolved Ca²⁺+CaM. Does CaM remain bound to the alpha subunit or CRD? Or is it also released? It is not blotted for in the pulldown, so I do not know where it is. The same with the alpha subunit. It feels incomplete to me, since you have 3 proteins but have only blotted for 1 of them (counting gamma or KD+CRD as one protein). The effect here is slight, so it's not strong evidence in the current state. I would expect your FLAG pulldown -Ca +TEV to be similar to the corresponding input. It seems like you're barely pulling down any KD, so it would be easy to lose that readout with a little washing. Also, why in the TEV treated samples is there full-length KD? Is it only partially being cleaved?

4) Where did the apo-B structure (Fig 5e) come from? I think the cryo-EM methods describe one sample without Ca²⁺ or ADP and another with both Ca²⁺ and ADP being imaged. This would have two differences, making assigning the conformational change as a consequence only of ADP binding difficult. If it is another structure, the PDB code should be noted.

Minor

1) The figure captions are not very descriptive. In particular, sample sizes (n) and what error bars represent should be described.

2) "Significant" was used to describe the result of Fig 5C but no statistical analysis was performed to assess significance.

3) Nomenclature is confusing and makes reading difficult. This is due to the use of English letters and Greek letters for the same things. For example, the $\alpha\gamma\delta$ subcomplex is referred to as AGC in the text but both $\alpha\gamma\delta$ and AGC are used in the figures. This may be convention in the field but negatively impacts the readability.

Reviewer #1 (Remarks to the Author):

This is an amazing manuscript that completes a story that began over 70 years ago. At the same time it also opens the door for a new chapter in understanding the complex mechanism that controls the activity of this essential protein kinase that is so fundamental for metabolism. The manuscript is clearly written and the cryoEM results are solid. I have no technical concerns whatsoever with this manuscript. The writing is actually exemplary. It is obviously a very complex macromolecular complex that describes many levels of regulation, but the authors have distilled down the essential features in a very concise way. The regulation by calmodulin is especially intriguing and surprising. The evolutionary insights into the alpha subunit are also intriguing. Is there a missing substrate that is hydrolyzed? The allosteric ADP site is another intriguing discovery. Is this a novel ADP/ATP binding site? It is extremely unusual to have the phosphates buried in so many basic residues.

While phosphorylase kinase was the first kinase to be discovered by Krebs and Fischer in the 1950s, their work was built on the remarkable foundation that was laid by Gerti Cory and Arda Green in the 1940s. Louise Johnson was also one of my close personal friends as well as Gerry Carlson, so I have followed this structure very closely for many decades and appreciate fully how challenging it has been. Obviously one cannot understand any kinase fully until you have the complete complex with all the domains, and this structure shows why the earlier studies were incomplete. This cryoEM structure has been long awaited and intriguing because a number of different states have been captured. I will delve more deeply into the activation mechanism associated with kinase activation, but the model proposed seems quite reasonable. This paper should be published quickly and highlighted as an historical landmark!

Dr. Susan Taylor

We would like to express our gratitude to Dr. Taylor for the enthusiastic comments and insightful feedback. The evolutionary insights into the α - and β -subunits have piqued our interest as well, prompting us to investigate their potential glucoamylase activities. However, our *in vitro* testing of both the $\alpha\gamma\delta$ subcomplex and the PhK holoenzyme did not reveal any detectable glucoamylase activity when tested against maltose, isomaltose, or glycogen, as depicted in Fig. 2e. It is possible that these are not suitable substrates, and the physiological significance of the glucoamylase-like domains in the α - and β -subunits warrant further investigation, particularly with regards to D1 α , which appears to possess a functional active site.

The presence of an allosteric ADP site in the β -subunit is indeed unusual.

Interestingly, a recent study has shown that ADP serves as an allosteric activator for eEF-2K, and is similarly situated in a pocket surrounded by positively-charged residues (doi: 10.1073/pnas.2300902120). Our previous research has demonstrated that ATP acts as an allosteric regulator of the pseudokinase Fam20A (doi: 10.7554/eLife.23990). The potential moonlighting functions of ADP/ATP as small molecule allosteric regulators, rather than simply as energy and phosphate donors, also merit further exploration in future studies.

We appreciate Dr. Taylor's thorough review of the field's history, and have incorporated some of these historical aspects into the Discussion section of the revised manuscript.

Reviewer #2 (Remarks to the Author):

This is a well written article that expands our knowledge on the sub-unit organisation and structure of phosphorylase kinase. The work is a nice extension of previous studies on the crystal and EM structures of this enzyme. While the new data is compelling and well conducted, I am afraid that I consider this to be an incremental advance in the field.

We thank the reviewer for acknowledging that our article is well written, and our data is compelling and well conducted.

Reviewer #3 (Remarks to the Author):

In this manuscript, Yang, Zhu et al. report the structure of the human muscle phosphorylase kinase complex. The complex is formed from a dimer of dimers in which the beta subunits form the core and the gamma and delta (kinase and calmodulin, respectively) subunits on the periphery. The authors describe key observed interactions and protein-protein interaction interfaces. Ca-free CaM molecules are bound to the C-tail of the gamma subunits in a region that is near an autoinhibitory domain. Upon addition of Ca²⁺, the KD and CaM are no longer observed in cryo-EM reconstructions, suggesting that separation of the KD from the autoinhibitory domain by CaM upon binding Ca²⁺ is the mechanism for kinase activation. While limited on biochemistry, this is a high-quality structural paper with clear figures for a physiologically important enzyme. However, I feel the proposed spring-loaded mechanism of activation needs additional validation. It is also difficult to understand the significance and what advancement to the field this research offers without a dedicated discussion section.

We appreciate the positive feedback from the reviewer. In response to these suggestions, we have further consolidated our biochemical experiments and have included a dedicated discussion section in the revised manuscript.

1) Are the farnesyl groups needed for formation of the complex? What effect does mutation of the Cys in the CaaX box have?

As per the suggestion, we have conducted this experiment. The results, as depicted in Fig. 3d, demonstrate that mutating Cys1090 β to Ala, but not Cys1220 α , leads to a shift of the PhK holoenzyme to a lower molecular weight position on size-exclusion chromatography. This finding aligns with our structural analyses, as the farnesyl groups on Cys1090 β are embedded in the β - β dimer interfaces (Fig. 3a) and play crucial roles in maintaining the structural integrity of the β_4 homotetramer, consequently influencing the PhK $\alpha_4\beta_4\gamma_4\delta_4$ hexadecamer. In contrast, the farnesyl group on Cys1220 α is involved in interacting with the γ -subunit (Fig. 4c). Given the extensive nature of the $\alpha\gamma$ interface, which involves multiple regions, it is not surprising that the Cys1220 α to Ala mutation does not have a significant effect on the complex formation.

2) Regarding activation of PhK by Ca²⁺, it is stated that the “protein sample was intact” but references a prior size exclusion run (Fig. 1b) that is not in the presence of Ca²⁺. How are you sure that CaM is still attached but is not observed due to

flexibility? I don't think the Superose6 chromatogram for that sample is shown. My concern is that dissociation of CaM could yield the same result (no observed density for CaM or the KD), so you have to provide evidence that the complex is still intact.

In the revised manuscript, we have provided the chromatogram and SDS-PAGE for the Ca²⁺-treated sample in Supplementary Figure 3c, where the band of calmodulin is clearly visible in the SDS-PAGE. Additionally, as mentioned in the paper, we generated a truncation mutant of the γ -subunit consisting of residues 1–326 and demonstrated its formation of a tight complex with calmodulin ($\gamma_{326}\delta$, Supplementary Fig. 3b). Based on these results, we have concluded that calmodulin remains attached to PhK in the presence of Ca²⁺.

3) I do not think that the cleavage experiment shown in Fig. 5C is strong evidence of the mechanism you are proposing. A cartoon in the main text may help to visualize your construct rather than just a supplemental sequence alignment. I think you have created KD-TEV site-CRD-AID-FLAG +CaM +alpha complex. Your assumption is that with Ca²⁺, the kinase domain will be released if the TEV site is cleaved. This seems reasonable based on your Ca²⁺ structure, but this structure also does not have a resolved Ca²⁺+CaM. Does CaM remain bound to the alpha subunit or CRD? Or is it also released? It is not blotted for in the pulldown, so I do not know where it is. The same with the alpha subunit. It feels incomplete to me, since you have 3 proteins but have only blotted for 1 of them (counting gamma or KD+CRD as one protein). The effect here is slight, so it's not strong evidence in the current state. I would expect your FLAG pulldown -Ca +TEV to be similar to the corresponding input. It seems like you're barely pulling down any KD, so it would be easy to lose that readout with a little washing. Also, why in the TEV treated samples is there full-length KD? Is it only partially being cleaved?

The cleavage of our original construct (KD-TEV site- α J-AID), in which residues 293–299 of the γ -subunit, located just before the α J helix, were replaced with the TEV protease recognition sequence, was suboptimal, resulting in inefficient cleavage. To enhance the efficiency of the biochemical assay, we designed another construct by inserting the TEV protease cleavage site between α J and AID (KD- α J-TEV site-AID). A corresponding cartoon was included for illustration (Fig. 7c), and this modification allowed for more effective TEV cleavage. Subsequently, we conducted a pull-down experiment using the twin-strep tag on the α -subunit, and the results were analyzed by Coomassie blue staining, instead of western blot.

As depicted in Fig. 7c, prior to TEV cleavage, the intact γ -subunit and calmodulin

were tightly associated with the α -subunit (left panel), irrespective of the presence of EDTA or Ca^{2+} . Even after the separation of KD from AID by TEV protease, KD remained bound to the α -subunit in the presence of EDTA (right panel). However, in the presence of Ca^{2+} , KD was not pulled down by the α -subunit after TEV cleavage, and only AID remained associated with the α -subunit. Calmodulin appeared to be less tightly associated with KD after TEV cleavage in the presence of EDTA. This observation may be attributed to the combination of EDTA and the disrupted end of α J, as calmodulin consistently exhibited tight binding to the intact γ -subunit even in the presence of EDTA (left panel), and the γ_{326} truncation mutant in the presence of Ca^{2+} , as evidenced in the $\gamma_{326}\delta$ sample (Supplementary Fig. 3b).

Taken together, our structural and biochemical data collectively support the spring-loaded mechanism we have proposed: in the absence of Ca^{2+} , calmodulin binds to the γ -subunit in a compact conformation; upon sensing Ca^{2+} , calmodulin undergoes a conformational change, facilitating detachment of KD and thereby de-inhibiting kinase activity.

4) Where did the apo-B structure (Fig 5e) come from? I think the cryo-EM methods describe one sample without Ca^{2+} or ADP and another with both Ca^{2+} and ADP being imaged. This would have two differences, making assigning the conformational change as a consequence only of ADP binding difficult. If it is another structure, the PDB code should be noted.

The apo- β structure originates from the inactive PhK structure. Given that the β -subunit lacks a Ca^{2+} binding site, we attribute the observed conformational changes to the binding of ADP, which exhibits clear density and interacts extensively with residues in the D1 β , D2 β , and D4 β domains. This interpretation is further supported by previous findings demonstrating that ADP can allosterically enhance PhK activity by binding to the β -subunit (Ref. 37).

Minor

1) The figure captions are not very descriptive. In particular, sample sizes (n) and what error bars represent should be described.

We have expanded the corresponding figure captions and included these information.

2) “Significant” was used to describe the result of Fig 5C but no statistical analysis was performed to assess significance.

We have redesigned this experiment as described above, and the results were analyzed by Coomassie blue staining, instead of western blot, which provides clearer visualization of the outcomes. Additionally, we have rewritten this section for improved clarity and accuracy.

3) Nomenclature is confusing and makes reading difficult. This is due to the use of English letters and Greek letters for the same things. For example, the $\alpha\gamma\delta$ subcomplex is referred to as AGC in the text but both $\alpha\gamma\delta$ and AGC are used in the figures. This may be convention in the field but negatively impacts the readability.

We have consistently utilized the Greek letter nomenclature system in the revised manuscript. Specifically, AGC has been changed to $\alpha\gamma\delta$, and GC to $\gamma_{326}\delta$.

Reviewer #3 (Remarks to the Author):

I thank the authors for performing extra experiments to support their manuscript. I think the changes have enhanced the story, and I do not have any additional comments.